neuroscience, biophysics, physiology

vision, opponency, retina, photoreceptor, Nymphalidae

**Author for correspondence:**
Primož Pirih
e-mail: primoz.pirih@bf.uni-lj.si

# Red-green opponency in the long visual fibre photoreceptors of brushfoot butterflies (Nymphalidae)

Gregor Belušič[1], Marko Ilić[1], Andrej Meglič[2] and Primož Pirih[1]

[1]Biotechnical Faculty, University of Ljubljana, Večna pot 111, 1000 Ljubljana, Slovenia
[2]Eye Hospital, University Medical Centre, Grablovičeva 46, 1000 Ljubljana, Slovenia

GB, 0000-0003-3571-1948; MI, 0000-0001-9910-0359; AM, 0000-0003-3837-8108; PP, 0000-0003-1710-444X

In many butterflies, the ancestral trichromatic insect colour vision, based on UV-, blue- and green-sensitive photoreceptors, is extended with red-sensitive cells. Physiological evidence for red receptors has been missing in nymphalid butterflies, although some species can discriminate red hues well. In eight species from genera *Archaeoprepona, Argynnis, Charaxes, Danaus, Melitaea, Morpho, Heliconius* and *Speyeria*, we found a novel class of green-sensitive photoreceptors that have hyperpolarizing responses to stimulation with red light. These green-positive, red-negative (G+R–) cells are allocated to positions R1/2, normally occupied by UV and blue-sensitive cells. Spectral sensitivity, polarization sensitivity and temporal dynamics suggest that the red opponent units (R–) are the basal photoreceptors R9, interacting with R1/2 in the same ommatidia via direct inhibitory synapses. We found the G+R– cells exclusively in butterflies with red-shining ommatidia, which contain longitudinal screening pigments. The implementation of the red colour channel with R9 is different from pierid and papilionid butterflies, where cells R5–8 are the red receptors. The nymphalid red-green opponent channel and the potential for tetrachromacy seem to have been switched on several times during evolution, balancing between the cost of neural processing and the value of extended colour information.

## 1. Introduction

A body of molecular, physiological and behavioural evidence gathered from diverse butterfly species is suggesting that butterflies possess good colour vision capabilities [1]. The ancestral trichromatic retinal architecture is based upon UV-, blue- and green-sensitive photoreceptors and allows for colour discrimination from UV to green, but not in the red colour range [2]. In pierid and papilionid butterflies, the colour vision range is extended with red-sensitive cells peaking beyond 600 nm, enabling tetrachromatic vision [1–3]. Physiological evidence for red-sensitive photoreceptors in the largest butterfly family Nymphalidae has so far been missing; a single study reported red-sensitive interneurons in *Heliconius* [4]. While some brushfoot butterflies retain the ancestral trichromatic plan [5–9], behavioural evidence for colour discrimination in the red wavelength range suggests the presence of a functional red colour channel in some species such as *Danaus plexippus* or *Heliconius erato* [10,11]. In this study, we screened diverse groups of nymphalid butterflies for physiological proof of red-sensitive photoreceptors.

Each optical unit of the butterfly compound eye, the ommatidium, contains nine photoreceptors [1]. Their microvilli fuse to a common light guide, the rhabdom. Photoreceptors R1 and R2 usually express UV- or blue-peaking opsins and contribute vertical (dorsoventrally oriented) microvilli to the distal rhabdom (figure 1*a*). Photoreceptors R3–8 express long-wavelength (LW)-sensitive opsins. In nymphalids, they contribute horizontal (R3&4) and diagonal (R5–8)

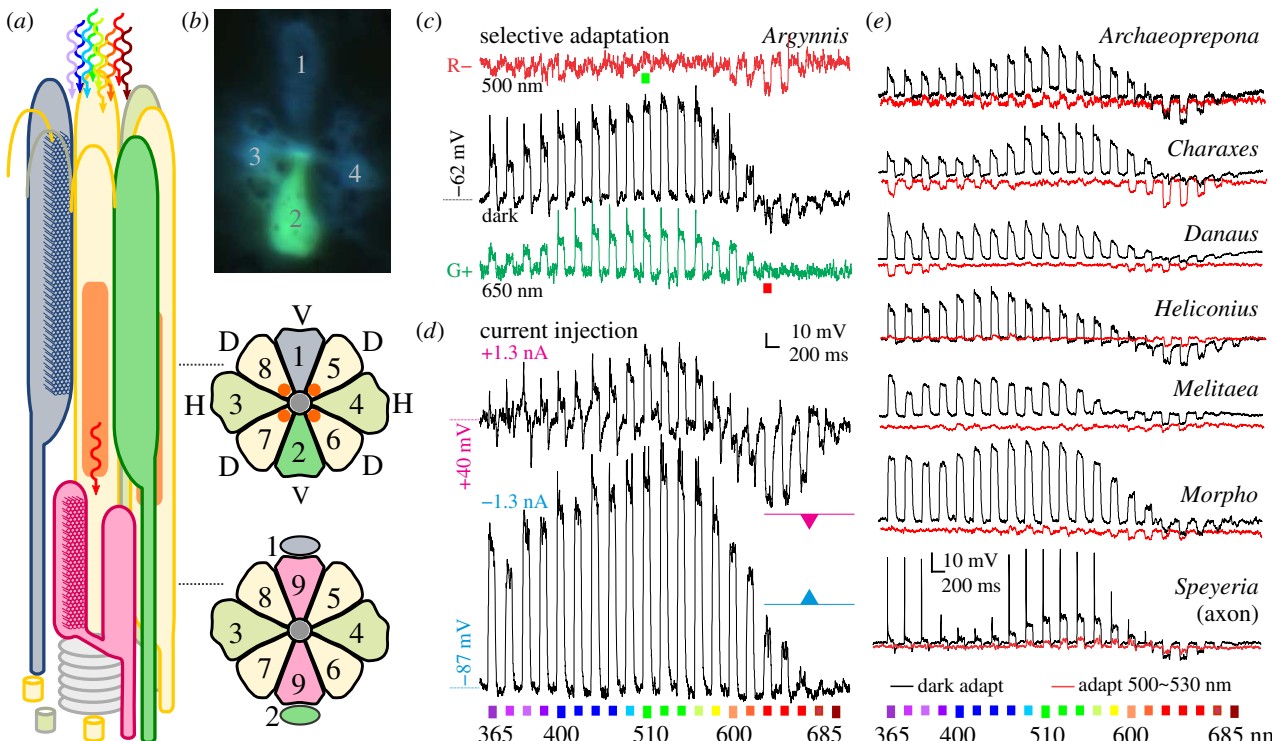

**Figure 1.** (*a*) Scheme of a nymphalid ommatidium (*not to scale*). Nine photoreceptors contribute microvilli to a fused rhabdom (*grey centre* in cross-sections). Cells R1&R2 (green and grey) have microvilli distally, basal R9 is bilobed. Filtering pigment (*orange*) in cells R5–8 (*yellow*) is apposed to the rhabdom. Light is reflected from the basal tapetum (*grey stack*), causing the eyeshine. (*b*) G+R− photoreceptor R2, loaded with dye. (*c*) Voltage responses of a dark-adapted G+R− cell to a spectral sequence of flashes (*black*). The G+ response (*green*) and R− response (*red*) are isolated with adapting light (650 and 500 nm, respectively). (*d*) Responses to red flashes are amplified with depolarizing current (magenta triangle indicates hyperpolarization in the red) and reversed with hyperpolarizing current (cyan triangle indicates depolarization in the red). (*e*) Voltage responses of G+R− cells in seven species, dark-adapted (*black*) and green-adapted (*red*), to a spectral sequence of flashes. (*b*,*c*,*d*) *Argynnis paphia*. (Online version in colour.)

microvilli along the whole rhabdom length [8]. The minute basal R9 has vertical microvilli and expresses a LW opsin [7,12]. Photoreceptors R1, R2 and R9 are the long visual fibres (LVF), projecting axons to the medulla, the second optical neuropil where colour information is being processed. The axons of photoreceptors R3–8 or short visual fibres (SVF) terminate in the first optical neuropil, the lamina [1]. It is not yet known how the neural signal is conveyed from R3–8 to the medulla, and how it contributes to colour vision. In *Papilio*, for instance, the only described pathway connecting the LW sensitive R3-8 with medulla is relaying in the lamina via large monopolar cells (LMC), neurons with very broad spectral sensitivities.

Spectral sensitivity of photoreceptors is determined by the expressed visual pigments, through direct inter-photoreceptor inhibitory synapses [13–16] and by optical filtering [1,2,17]. The downwelling light is filtered by the visual pigments in their rhodopsin and metarhodopsin states. A red screening pigment [1,10,17] (figures 1*a* and 2*e*,*f*) can further modify the spectral composition of downwelling light, resulting in red-shifted sensitivity spectra of proximal cells.

Many brushfoot butterflies have ommatidia with a red screening pigment closely apposed to the rhabdom [10]. In these ommatidia, photoreceptors with red-shifted sensitivity, can be expected in the proximal retina. Here, we show that in the species with red ommatidia, the retina contains green-sensitive and red-sensitive photoreceptors with LVF that together build opponent pairs, most likely via direct, inter-photoreceptor synapses. These opponent pairs are the probable retinal substrate for colour discrimination in the orange-red part of the spectrum, offering potential for tetrachromacy.

## 2. Results

We have examined the retinae of 10 nymphalid species by intracellular electrophysiological measurements from single photoreceptor cells, using stimulation with narrow-band spectral light, selective adaptation and current injection. We found photoreceptors that depolarized when stimulated with monochromatic flashes ranging from the UV to green and hyperpolarized to red flashes (figure 1*c*,*e*). We assumed that the hyperpolarizations were caused by inhibitory synapses from red-sensitive photoreceptors. The hyperpolarizing responses could be isolated with green (G) adapting light and silenced with red (R) adapting light (figures 1*c*,*e*, 2*a–c*, 3*a–c*). When recording from the axons, the responses to red light could be amplified by injecting a depolarizing current and sign-reversed by injecting a hyperpolarizing current (figure 1*d*; electronic supplementary material, figure S1). The responses to red light had a reversal potential of about −70 mV. This suggests that chloride current, likely conducted through histaminergic channels, is involved in hyperpolarization [15]. We termed the novel photoreceptor class G+R−: green-sensitive cells (G+) inhibited by postsynaptic currents from red-sensitive units (R−). Below, we provide the evidence that the G+R− cells are R1&2 and their R− units are likely R9. The novel G+R− photoreceptor class is the retinal basis for green-red colour opponency.

Detailed spectral sensitivity measurements using selective adaptation showed that the G+ cells and R− units peaked at approximately 520 nm and approximately 620 nm, respectively (figure 2*a–c* and table 1). The intensity–response functions, recorded with green and red stimulation in dark-

*Proc. R. Soc. B* **288**: 20211560

**3**

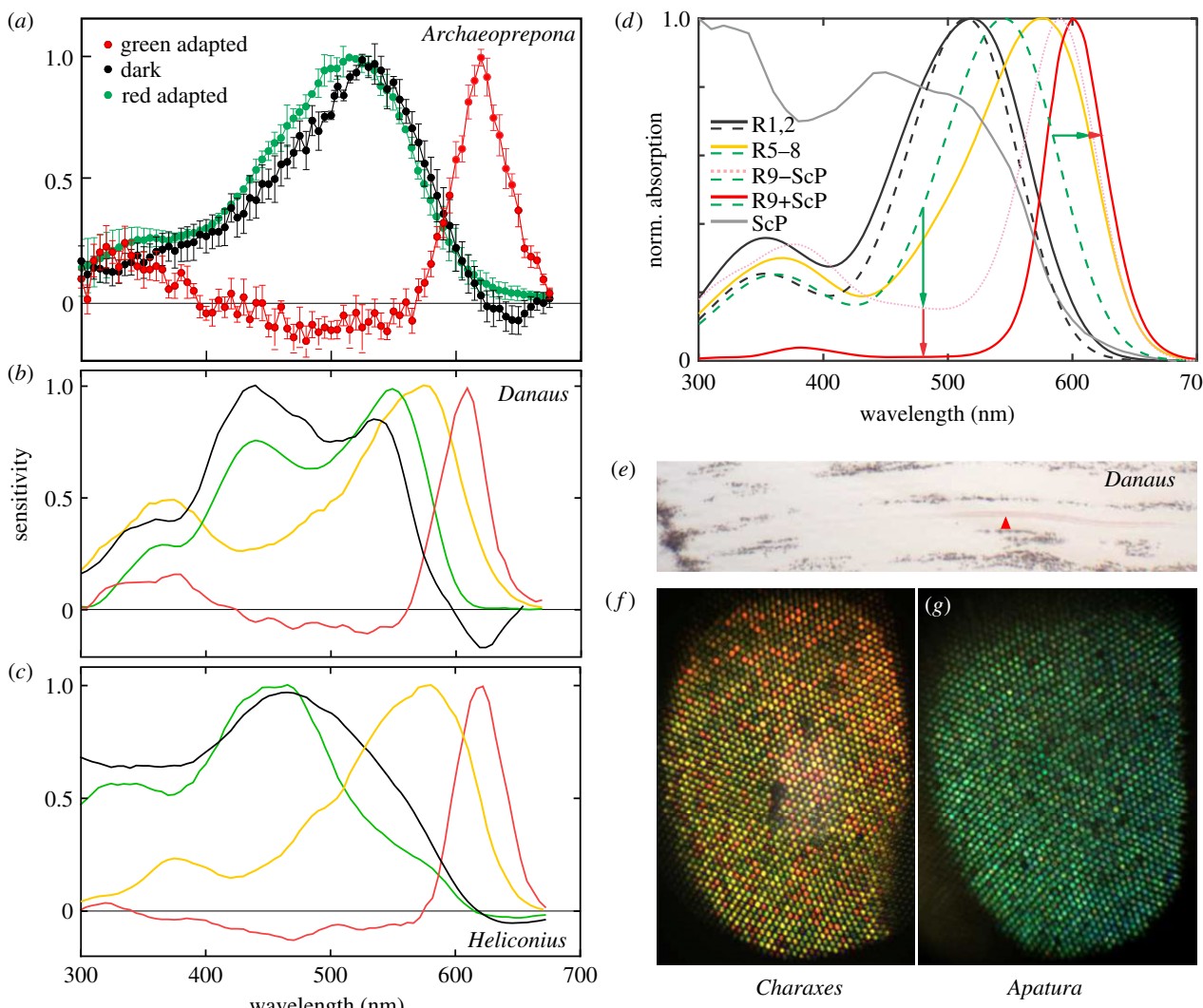

**Figure 2.** (*a*) Normalized spectral sensitivity (m ± s.e.) of the G+R− cells (*black*), R− units (*red*) and G+ units (*green*) (all adapted states: $n = 2$; dark-adapted and green-adapted: $n = 4$; only green-adapted: $n = 5$). (*b*) Spectral sensitivities of a single G+R− (dark-adapted state, adapted with 525 nm and 650 nm light: *black, red, green traces*), a single diagonal cell (*yellow trace*; PS shown in figure 3*d*) in *Danaus plexippus*. (*c*) Idem in *Heliconius erato*. (*d*) Modelled spectral sensitivity of G+R1/2 (*black*), R5–8 (*yellow*) and R9 for the case with the red pigment present (*red trace*) and absent (*pink dotted trace*); templates for visual pigments Rh515 (*dashed black*), Rh545 (*dashed green*). Measured absorbance of red screening pigment (*grey*). Sensitivities of R5–8 and R9 are shifted due to filtering with opsins (*green arrows*) and screening pigment (*red arrows*). (*e*) Unstained longitudinal section of the retina with the red pigment (*arrow*) apposed to the rhabdom. (*f*) Eyeshine of the two-tailed pasha, showing the mosaic of green-, yellow- and red-reflecting ommatidia. (*g*) Eyeshine of the lesser purple emperor, showing a uniform pattern of green reflecting ommatidia. (Online version in colour.)

and green-adapted state, respectively, indicated that the hyperpolarizing R− unit had lower light sensitivity and a smaller dynamic range than the G+ unit. The parameters of the sigmoid function, fitted to the intensity–response curve, were $\log_{10}$ of light intensity evoking half-maximal response $R_{G+} = -2.51$, $R_{R−} -1.21$; slope $n_{G+} = 0.86$, $n_{R−} = 1.05$ (figure 3*a*).

Response to a rotating linear polarizer was measured to estimate the angular maximum of polarization sensitivity (PS), which coincides with the microvillar orientation, indicative of the receptor position within the ommatidium [18]. The magnitude ($\Psi$) and angular maximum ($\Phi$) of PS were measured in dark-adapted and green-adapted cells. PS magnitude was low for the G+ unit ($\Psi_{G+} = 1.2 \pm 0.2$) and modest for the R− unit ($\Psi_{R−} = 2.0 \pm 0.9$). The angular maxima of both units ($\Phi_{G+} = 103 \pm 5°$, $\Phi_{R−} = 96 \pm 13°$) were consistent with the vertical orientation of microvilli (figures 1*a*, and 3*b,c*). Micro-electrode dye injection confirmed that G+R− cells were indeed R1/2 (figure 1*b*; electronic supplementary material,

figure S3). In *D. plexippus*, *H. erato* and *M. athalia* some G+ R− R1/2 cells had broadened sensitivity with two maxima, in the green and blue (figure 2*b,c*; electronic supplementary material, figure S2), in line with the recently found co-expression of blue and LW opsins in *Heliconius* R1&2 [19]. Varying levels of opsin co-expression may be the main cause for the different spectral sensitivities of G+R− cells in different species (electronic supplementary material, figure S2).

Within 478 recorded cells (electronic supplementary material, table S1), we found all common photoreceptor classes (UV, B, G), including green-sensitive cells with spectral sensitivity maxima shifted to orange (figure 2*b,c*) and with diagonal PS maxima (figure 3*d*), likely R5–8, but we never recorded directly from a red receptor. We posit that R− units are the minute R9 cells, inaccessible for direct microelectrode impalement. Their functional properties are consistent with a short rhabdomere being screened by the distal filtering and visual pigments. Receptive fields of the green and red units overlapped, showing that the signals

Proc. R. Soc. B 288: 20211560

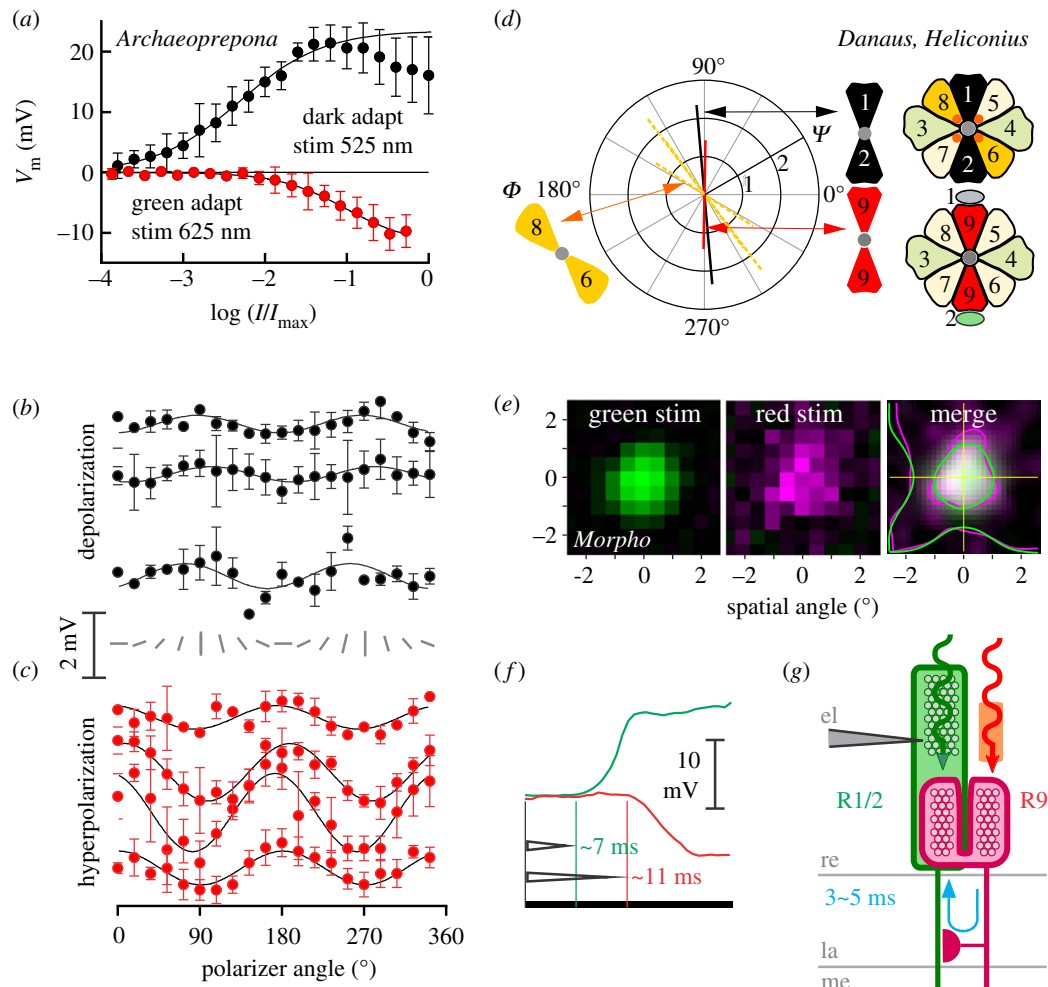

**Figure 3.** (a) Stimulus intensity–response dependence of G+R− cells, fitted with sigmoid functions (m ± s.e., N = 4). Dark-adapted and measured at 525 nm (*black*; responses > −1 log were excluded from the fit), green-adapted (500 nm) and measured at 625 nm (*red*). (b) PS of dark-adapted G+R− cells measured with 525 nm stimuli. (c) PS of green-adapted cells measured with 625 nm stimuli. Rotating bars indicate polarizer orientation. Fits to voltage responses from 3 to 6 rotations of the polarizer in one cell. (d) The angle of maximal PS Φ versus PS ratio Ψ (bar length) for G+ unit (*black*), R− unit (*red*) and diagonal receptors (*orange*) from figure 2b,c. (e) Normalized voltage responses of a G+R− cell, spatially stimulated by 0.5 × 0.5° green or red squares. The merged image shows interpolated data, overlaid with half-maximal response isolines and central transects. (f) Voltage responses of G+R− to green and red flashes have different delays. (g) Proposed functional scheme of G+R− opponent pair with axonal propagation and synaptic delay (*blue arrow*). Electrode (*el*); retina (*re*); lamina (*la*); medulla (*me*). (Online version in colour.)

from G+ and R− units originate in a single ommatidium (figure 3e). The hyperpolarizing responses are delayed (figure 3f; electronic supplementary material, figure S4) due to axonal propagation and synaptic delay, and possibly due to the red receptor's slower response kinetics (figure 3g).

In order to support the indirect electrophysiological evidence for the red unit being R9, we modelled a rhabdom with or without the red screening pigment, with R1–2 and R3–9 expressing rhodopsins R515 and R545, respectively [7,12]. The light transport model (figure 2d) shows that screening by rhodopsins and their metarhodopsins causes a red sensitivity shift and sensitivity broadening in photoreceptors R3–8, partially explaining the differences between the spectral sensitivity curves, obtained in the dark, and those obtained with a constant adapting light (figure 2a–c). The sensitivity peak of R9 is further LW shifted to red by the screening pigment (figure 2e). The model predicts that the absolute sensitivity of photoreceptors R9 is significantly lower (approx. 2 $\log_{10}$ units; *not shown*) than that of green R1/2, in line with the differences between the measured intensity–response curves of G+ cells and R− units (figure 3a). In ommatidia without the red screening pigment, R9 should

have substantial sensitivity in the green (figure 2d, pink dotted curve). We could not identify such units in our recordings. Importantly, in red ommatidia, the screening pigment significantly reduces the sensitivity of R9 in the green spectral range (figure 2d), in line with the measured spectral sensitivity of R− units, exhibiting a sharp red sensitivity peak and a small UV sideband (figure 2a–c; electronic supplementary material, figure S2b–d). In the red ommatidia, the two photoreceptors in the G+R− opponent pair have mutually exclusive, non-overlapping spectral sensitivities: G+ are insensitive to red stimuli due to opponent signalling from R−which are in turn insensitive to green stimuli due to the red screening pigment. The reduced overlap of spectral sensitivity likely enhances colour discrimination ability.

## 3. Discussion

Direct opponent interactions between the photoreceptors, similar to those observed in *P. xuthus* [15], seem to be a common feature of the butterfly retina. In all brushfoot butterflies studied here, we recorded opponent combinations,

**Table 1.** Spectral properties of G+R− cells in studied species.

| species | subfamily/tribe | R− $\lambda_{max}$ (nm) | R− FWHM (nm) | G+ $\lambda_{max}$ (nm) |
|---|---|---|---|---|
| *Archaeoprepona demophon* | Charaxinae/Preponiini | 620 | 47 | 530 |
| *Argynnis paphia* | Heliconiinae/Argynnini | 620 | 45 | 535 |
| *Charaxes jasius* | Charaxinae/Charaxini | 620 | 40 | 535 |
| *Danaus plexippus*[a] | Danainae/Danaini | 610 | 45 | 440 |
| | | | | 510 |
| *Heliconius erato*[a] | Heliconiinae/Heliconiini | 620 | 50 | 465 |
| | | | | 545 |
| *Melitaea athalia* | Nymphalinae/Melitaeini | 610 | 40 | 450 |
| *Morpho peleides* | Satyrinae/Morphini | 610 | 60 | 500 |
| *Speyeria aglaja* | Heliconiinae/Argynnini | 620 | 50 | 530 |
| *Apatura ilia* | Apaturinae/Apaturini | — | — | 530 |
| *Vanessa atalanta* | Nymphalinae/Nymphalini | — | — | 530 |

[a]Spectral sensitivity has two peaks, in the blue and in the green, hence two values for $\lambda_{max}$. FWHM, full-width half-maximum (bandwidth at 50% sensitivity).

formed by UV+G− and B+G− units (*not shown*). However, we only found G+R− cells in butterflies with red ommatidia (figures 1*e* and 2*f*; electronic supplementary material, figure S5) and not in butterflies that have uniform eyeshine (figure 2*g*). The presence of ommatidia with red screening pigments is not only correlated with the finding of G+R− cells, but seems to be a requirement for the red-shifted sensitivity of R9 and hence the implementation of the extended colour discrimination range. The red admiral (*Vanessa atalanta*) has uniform orange eyeshine [10], indicative of red screening pigment absence, and a limited ability for discrimination of red colours, while the red postman (*Heliconius erato*) has a mosaic with about 50% red-shining ommatidia, and a behaviourally confirmed extended colour discrimination range [7,10,17,19]. We did not find any UV- or blue-sensitive R1/2 cells receiving opponent signals from red-sensitive cells. R9 thus seems to have a specific role in the red ommatidia, in providing antagonistic input only to the green-sensitive R1/2 photoreceptors, while its role in the non-red ommatidia remains unknown. Still, the long fibre of R9 may play a more general role in colour processing in the medulla.

The red-green antagonism spectrally narrows the sensitivity of the green channel and is likely a functioning part in the implementation of tetrachromacy. The lower absolute sensitivity of R− units (figure 3*a*) suggests that red vision may be limited to bright conditions. The sensitivity spectra of R5–8 are broadband-green [7] or orange-shifted due to screening (figure 2*b,c*). While these SVFs might indirectly contribute to colour vision, we have not found R5–8 cells that would provide antagonistic input to the vertical green receptor.

In nymphalid butterflies with red-reflecting ommatidia, the red receptors may have been overlooked in the previous electrophysiological studies, [8,20] perhaps due to having non-optimal stimulus intensity and aperture, or due to ascribing the indirect, minute negative signals to an extracellular (electroretinogram) artefact [21]. We note that the sign of extracellular artefacts cannot be reversed by electrode current injection, used here.

R9-based red receptors have been directly recorded only in some Hymenoptera [22], but the study does not report whether the wasp's R9 (or main) photoreceptors had hyperpolarizing responses. The spectral sensitivity of the red receptor has been previously reported for a nymphalid *Anartia amathea*, via an optical measurement of pupil action using selective adaptation [23]. The reported sensitivity of the red receptor, peaking at approximately 610 nm, with a sideband below 420 nm, is consistent with our experiments and the model.

## 4. Conclusion

The novel LVF G+R− photoreceptor class is present in diverse groups of brushfoot butterflies, implemented via antagonism between a green-sensitive R1/2 and a red-sensitive R9. The telltale sign for the presence of G+R− cells are red-shining ommatidia. A similar implementation of the red colour channel might exist in Lycaenidae [24], while in Pieridae and Papilionidae, R5–8 have assumed the role of red receptors instead [1], and the role of R9 in these two families is not known. The red channel appears to be absent from nymphalid butterflies with uniform eyeshine. This suggests that the additional chromatic channel may be associated with a high build cost and additional energy demands for neural processing. Whether a brushfoot butterfly *goes red*, ultimately depends on the spectra of its visual interests.

## 5. Experimental methods

### (a) Animals

The butterfly species were chosen to represent six subfamilies of Nymphalidae: Danainae (monarch *Danaus plexippus*), Charaxinae (two-tailed pasha *Charaxes jasius*, prepona *Archaeoprepona demophon*), Morphinae (blue morpho *Morpho peleides*), Heliconiinae (red postman *Heliconius erato*, silverwashed fritillary *Argynnis paphia*, dark green fritillary *Speyeria* (formerly *Argynnis*) *aglaja*), Nymphalinae (heath fritillary *Melitaea athalia*, admiral *Vanessa atalanta*), Apaturinae (lesser purple emperor *Apatura ilia*). Butterflies were purchased as pupae from The Butterfly Farm-Costa Rica

Entomological Supply (*A. demophon, M. peleides, D. plexippus, H. erato*). Adults were collected around the Department (*A. paphia, S. aglaja, M. athalia, A. ilia*) and near Zadar, Croatia (*C. jasius, V. atalanta*). After eclosion, the butterflies were kept at 27°C and 80% relative humidity and regularly fed sucrose solution.

## (b) Electrophysiological recordings

The animals were immobilized in plastic tubes with a mixture of beeswax and resin and fixed with the head in the centre of rotation into a miniature goniometer. After the animal was pre-oriented for the recording, a small hole for the micro-electrode was cut into the cornea and sealed with silicon vacuum grease. The reference electrode was a 50 μm diameter Ag/AgCl wire, mounted on the miniature goniometer, inserted below the cornea of the recorded eye, in order to mini-mize extracellular (electroretinogram) artefacts. The mini goniometer was then fixed to a large goniometer, which additionally carried a piezo-driven micromanipulator (Sensa-pex, Oulu, Finland). Again, the eye was carefully positioned at the centre of rotation of the large goniometer. The dorsoven-tral axis of the compound eye was aligned with the $z$-axis of the recording microelectrode, yielding a maximal number of cell impalements and rendering all parts of the eye accessible for the recording, including the extreme dorsal and ventral regions. The location of the microelectrode tip during the pen-etration was determined with the micrometre dial on the $z$-axis. The depth of recording was determined by the location of the hole on the cornea, the electrode angle and estimated according to the relative quantities of impaled distal receptors R1&2 versus the proximal R3–8. The electrode trajectory was also visible in histological sections; current-clamp experiments were only successful at least 250 μm proximally from the cornea. Recordings from R1&2 axons were obtained in the proximal retina, where R1&2 do not have the rhabdomeres, in the fenestrated layer below the retina or in the lamina.

Single-cell recordings were performed using a high impe-dance amplifier (SEC-10LX, NPI, Tamm, Germany) in bridge mode or discontinuous clamp mode at 20 kHz and 0.25 duty cycle. The electrodes, pulled from borosilicate glass on a hori-zontal puller (P-2000, Sutter, Novato, USA), filled with 3 M KCl, had a resistance in the range 80–120 MΩ. The signals were acquired with a laboratory interface (Micro1401 mkII, Cambridge Electronic Design, Ltd. Cambridge, UK) con-trolled by software WinWCP, version 5.5.4.

Opponent signals could be manipulated with current injec-tion only in cells that were impaled in the axons. Some recorded LVFs exhibited spikelets (*Argynnis* in figure 1*c*, green trace; electronic supplementary material, figure S4*a*), indicating the presence of voltage-gated Na$^+$ or Ca$^{2+}$ channels facilitating propagation of graded signals along the long fibres. The spikelets became spikes when recording from the axons in the lamina (*Speyeria*, figure 1*e*). Many units also had larger dark noise (*not shown*), possibly due to tonic transmitter release from the presynaptic opponent cell. The current, required to reverse or amplify synaptic voltage, was larger when record-ing distally in the retina, remotely from the putative synaptic site. This led to extremely large voltage read-outs during switched current injections. Due to microelectrode rectification and pronounced space clamp problems in thin and elongated cells, we were unable to perform a systematic investigation of the reverse potential of synaptic currents.

To visualize the impaled cells, blunter microelectrodes (60–80 MΩ in 3 M KCl) were loaded with Lucifer yellow (L0144, Sigma-Aldrich, St Louis, USA) in 0.1 M LiCl and back-filled with 1 M LiCl, yielding higher electrode resistance (greater than 200 MΩ). After spectral identification, the impaled cells were injected with the dye by pumping hyper-polarizing current pulses (−1 nA) for approximately 15 min. The retina was then isolated, fixed, processed and observed as described in the Anatomy section.

## (c) Spectral stimulator

The light stimuli were provided by two sources. The first con-sisted of a 75 W xenon arc lamp (Cairn Research, Kent, UK), a monochromator (B&M Optik, Limburg, Germany), a shutter and a computer-controlled neutral density wedge filter (Thor-labs, Bergkirchen, Germany). The second source was a 'LED Synth', a wavelength combiner based upon LEDs and a dif-fraction grating [25]. The peak wavelengths of the LED synth channels were 365, 375, 390, 402, 423, 437, 452, 470, 495, 512, 525, 543, 560, 576, 592, 600, 619, 630, 660, 672 and 685 nm. Both sources were combined with a polka-dot beam splitter (Thorlabs). The intensity of both sources was measured with a radiometrically calibrated spectropho-tometer (Flame, Ocean Optics, USA) and adjusted to equal photon (isoquantal) flux density at all wavelengths (max. $1.5 \times 10^{15}$ photons cm$^{-2}$ s$^{-1}$). The two sources were aligned to yield coaxial illumination with the far-field aperture adjus-table in the range from 1.5° to 20°. The spectral sensitivity or stimulus intensity–response dependence of a cell was scanned with one source, while the other source was used for selective adaptation. For PS measurements, a UV-capable polariser (OUV2500, Knight Optical, UK) was placed into the monochromator beam and rotated for three to five full rotations. The flashes were applied each 18° step, while the unpolarized LED source was used to selectively adapt the cells. The intrinsic degree of polarization of both stimulator beams was less than 1%.

## (d) Spatial light stimulator

To map the spatial receptive fields of the spectrally character-ized photoreceptors, the animal was carefully rotated to face a back-projection screen (ST-Pro-X, Screen-Tech e.K., Hohe-naspe, Germany) so that the microelectrode remained in the impaled cell. Spatial stimuli were presented on a back-illumi-nated projection screen through an RGB DLP projector (LightCrafter 4500, Texas Instruments, USA) with approxi-mate emission peaks at 460, 530 and 620 nm. Monochrome spatial stimuli were presented at a refresh rate 220 Hz using the software package PsychoPy [26]. After the centre of the receptive field had been found using horizontal and vertical bars, the receptive field was mapped by presenting a flashing $0.5 \times 0.5°$ square running left to right, top to bottom, on a $10° \times 10°$ grid.

## (e) Analysis of electrophysiological data

The response amplitudes were measured as the difference between the resting membrane potential in the dark and the sustained light response after the initial peak. A sigmoid, $V = (V_0 \, I^n) \, (I^n + R^n)^{-1}$, was fitted to the measured intensity–response function $V(I)$. The parameter estimates were used for a reverse transformation of the response amplitudes to

isoquantal spectral stimuli, yielding (spectral or polarization) sensitivities. The second intensity–response function, obtained in G+R− cells by flashing red pulses over a constant green adapting background, was used to reverse transform the hyperpolarizing responses. The sensitivities are presented in the graphs with the larger of the two peaks normalized to +1. PS was measured as described previously [27]. The response oscillations to the rotating linear polarizer were fitted with a squared cosine function and parametrized as the angle of maximal sensitivity $\Phi$ and the polarization sensitivity ratio $\Psi$.

## (f) Anatomy

Isolated retinae were fixed for 3 h in 4% paraformaldehyde and 3.5% glutaraldehyde, dehydrated in ethanol series and then embedded in Spurr's resin. Semi-thin sections (1 μm) were cut on an Ultracut S ultramicrotome (Leica, Nussloch, Germany) with a diamond knife (Diatome, Nidau, Switzerland). Unstained sections or sections, stained with Azur II, were observed with an upright light microscope (Axioskop 2 FS, Zeiss, Oberkochen, Germany) with a multi-band fluorescence cube (no. 69401 and no. 89402, Chroma, Bellows Falls, USA), coupled to a multispectral LED fluorescence light source (Niji, Bluebox Optics, Blackwood, UK).

## (g) Hyperspectral imaging of red pigments

The absorbance spectrum of the peri-rhabdomal screening pigment was obtained from an unstained longitudinal eye section of *Danaus plexippus*. An upright microscope (Axioskop 1 FS, Zeiss, Oberkochen, Germany) with a custom motorized z-axis stage, a Zeiss Ultrafluar 40 × NA0.60 immersion objective (part 440015) and an achromatic condenser (NA 0.9) was coupled to a light source (75 W XBO lamp with a motorized monochromator, Deltascan 4000, PTI/Horiba). Hyperspectral image stacks were acquired with a CMOS monochrome camera (Blackfly BF-U3-23S6M, FLIR). Prior to taking the hyperspectral stacks, the best focus position at each wavelength was determined using a calibration slide. Image registration and analysis were performed in Fiji [28], using the StackReg plugin [29]. Further details are available elsewhere [30].

## (h) Eyeshine

Eyeshine images were taken with a custom epi-illumination microscope with a telescopic tube, using a Zeiss LD Epiplan 20 × NA0.40 objective and an CMOS RGB camera (Blackfly S BFS-U3-200S6M, FLIR). Optical design and measurement principles are described in detail elsewhere [9,17].

## 6. Light transport model

The absorption in the rhabdom was modelled with longitudinal light transport across slabs [3,9,31]. The model ignored the effects of distal optics, diffraction, waveguide modes and birefringence. The rhabdom length was 450 μm and the slab length was d$z$ = 5 μm. The model was implemented in GNU Octave [32]. The four-dimensional arrays were set in the order wavelength $\lambda$, receptor index $r$, depth $z$, time $t$, with the ranges $\lambda$ = [300:700] nm and $z$ = [0:5:450] μm and were processed using broadcast operators over singleton dimensions.

Govardovskii templates $\Gamma(\lambda)$ were used for rhodopsin (R) and metarhodopsin (M) pigment isoforms [33]. Templates were set according to electrophysiologically measured sensitivity maxima of R1–8. Green receptors R1/2 were modelled with the isoform pair R515/M495, while the main pigment isoform pair was set to R545/M505. The peak absorption coefficient was set to $\alpha_R$ = 6 mm$^{-1}$ and $\alpha_M$ = 7.5 mm$^{-1}$. The red pigment template was taken from the hyperspectral measurement (figure 2$d$). The rhabdom cross-section area fraction $\rho_{zr}$ of photoreceptors R1–R9 in each slab was interpolated and modified from the morphological data from the chestnut tiger, *Parantica sita*, a large nymphalid butterfly from the subfamily Danainae [8]. The basal receptor R9 contributed to the rhabdom between 400 and 450 μm.

For each slab ($z$), photoreceptor ($r$ = 1, …, 9) and pigment isoform index ($p$ = R,M), effective absorption coefficients, $\kappa_{zrp} = \alpha_p \, \rho_{zr} \, f_{zp}$, were calculated using the peak absorption coefficient, $\alpha_p$, rhabdom cross-section fraction, $\rho_{zr}$, and the pigment isoform fraction, $f_{zp}$, constrained to ($f_R + f_M = 1$). Red screening pigment was present between 230 and 420 μm and technically implemented as two identical isoform pigments in an additional cell with negligible $\rho_z$. Its peak absorption coefficient $\alpha_S$ was set to approximately match the exiting spectral flux with the eyeshine reflectance spectra measured from larger butterflies with red ommatidia (e.g. *Morpho, Danaus; not shown*), or set to zero for the model without red screening pigment.

The starting downward flux at the cornea was isoquantal between 300 and 700 nm (*white supercontinuum*). The initial pigment isoform fractions were set to the dark-adapted state ($f_R$ = 1, $f_M$ = 0). The downwelling spectral flux exiting each slab with thickness d$z$ was calculated as $I_{out} = I_{in}$ exp $[-dz \, \Sigma_r \, \Sigma_p \, (\kappa_{zrp} \, \Gamma_{rp})\,]$. The light reaching the tapetum was fully reflected and the procedure repeated upwards. The downwelling and upwelling flux in each slab were summed to obtain the bidirectional (actinic) flux $I_z$. The photochemical reaction rates $k_{zrp} = \int (\kappa_{zrp} \, \Gamma_{rp} \, I_z) \, d\lambda$ were used to calculate the rhodopsin fraction change at each slab as d$f_R$ = ($k_M \, f_M - k_R \, f_R$) when solving the system as an ODE problem, or as $f_{\infty R} = k_M / (k_R + k_M)$ when calculating the equilibrium. In the latter case, the equilibrium was reached in 5–10 iterations due to the interdependence of the pigment isoform fractions $f_\infty$ and the light flux $I_z$. The two calculation methods reached numerically similar equilibria. In post-processing, the effective light-adapted spectral sensitivity of each of the nine photoreceptors $Q_{\infty r}$ was calculated by summing absorption by the rhodopsin isoform over z-slabs, $Q_{rR} = \Sigma_z \, (I_z \, \rho_{zr} \, \kappa_{zR})$. The equilibrium fraction of the main rhodopsin $f_{\infty R545}$ was approximately 0.45 distally, declining to $f_{\infty R545} \approx 0.15$ basally. The effective peak sensitivity of R3–8 shifted from approximately 545 nm distally to above 600 nm basally.

Data accessibility. The data are available from the Dryad Digital Repository: https://doi.org/10.5061/dryad.1vhhmgqtc [34] and Zenodo: https://zenodo.org/record/5207322#.YS5V5jqxVaQ [35]. The data are provided in the electronic supplementary material [36].

Authors' contributions. G.B.: conceptualization, data curation, formal analysis, funding acquisition, investigation, methodology, project administration, resources, supervision, validation, visualization, writing-original draft, writing-review and editing; M.I.: formal analysis, investigation and methodology, software; A.M.: investigation and methodology; P.P.: conceptualization, data curation, formal analysis, funding acquisition, investigation, methodology, project

administration, resources, software, supervision, validation, visualization, writing-original draft, writing-review and editing. All authors gave final approval for publication and agreed to be held accountable for the work performed therein.

Competing interests. The authors declare no competing interests.

Funding. This study was financially supported by the AFOSR/EOARD (grant no. FA9550-15-1-0068, to G.B. and M.I.), by the Slovenian Research Agency (grant no. P3-0333 to G.B., A.M. and P.P.) and jointly by the European Regional Development Fund and MESS of Slovenia (grant no. 5442-1/2018/434 to P.P.).

Acknowledgements. We thank Votan d.o.o. for the technical development of LED synthesizers. We are grateful to Drs Žiga Fišer and Rudi Verovnik for collection and determination of butterflies. We thank Drs Adam Blake, Mathias Wernet and Bodo Wilts for their valuable comments on the previous version of the manuscript and the anonymous reviewers for their insightful comments.

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
