## [Peer Review File · Proceedings of the Royal Society B: Biological Sciences]

Review History

RSPB-2021-1560.R0 (Original submission)

Review form: Reviewer 1

Recommendation

Accept with minor revision (please list in comments)

Scientific importance: Is the manuscript an original and important contribution to its field?

Excellent

General interest: Is the paper of sufficient general interest?

Excellent

Quality of the paper: Is the overall quality of the paper suitable?

Excellent

Is the length of the paper justified?

Yes

Should the paper be seen by a specialist statistical reviewer?

No

Do you have any concerns about statistical analyses in this paper? If so, please specify them explicitly in your report.

No

It is a condition of publication that authors make their supporting data, code and materials available - either as supplementary material or hosted in an external repository. Please rate, if applicable, the supporting data on the following criteria.

Is it accessible?

Yes

Is it clear?

N/A

Is it adequate?

N/A

Do you have any ethical concerns with this paper?

No

Comments to the Author

This manuscript clarifies a long-standing important question in insect colour vision: the potential role of the mystic photoreceptor number 9. The authors investigated this in several species of Nymphalid butterflies, but it is likely relevant beyond this group. The methods are all sound and the results clear, but in some parts, the description is a bit too short for a general reader to understand all details and the impact of them.

This is mainly a result of the short format, but could also be helped by clearer definitions of the exact receptor types and units in focus. I have made a few suggestions for such clarifications, all very minor.

I also think it would be nice to cite the earliest (and since then, often disputed) finding of electrical inhibition in butterfly photoreceptor signals: Matic T (1983) Electrical inhibition in the retina of the butterfly *Papilio I*. Four types of photoreceptors. *J Comp Physiol* 152: 169-82, and mention the more recent findings of mutual photoreceptor inhibition in the fly retina, there are various recent papers on *Drosophila*.

Else, I think this is a very important contribution to the field, congratulations to the authors.

Minor comments:

Title and elsewhere: not everybody knows that the wonderful name brushfoot butterflies refers to Nymphalidae, maybe good to mention it as "(Nymphalidae)" in the title

Line 53: replace "how does it" with "how it does"

Line 55f: it would be good to better relate the sentence on filtering with the "optical filtering" in the sentence before, maybe by moving "optical filtering" last in the list.

Lines 67-77: this is a very short summary of results, hard to digest for a non-specialist reader. For instance, while it can be inferred from the fact that red light induces hyperpolarization, this is not clear from a non-specialist reader, so "the inhibition" mentioned in line 76 may need to be introduced more explicitly. A few additional points that could be clarified:

Line 67: "measurements" sounds very general, maybe good to add "electrophysiological" to clarify

Line 68: "spectral stimulation" means stimulation with spectral light" - maybe good to say, for the non-specialists reader

Line 77: not clear what a "receivingfrom colour-opponent, red-sensitive unit" means, does it

rather mean: receiving colour-opponent...from red-sensitive units”?

Line 78ff: you have just defined “R+G-“ class of receptors, now you introduce “G+ and R- units“ – maybe better to use consistent naming? Are you now talking about one cell type now, or about two different units that give input to the G+R- cells?

Line 88: to clarify which cells you record from, is it possible to give the information that G+R-calls are R1/2 a bit earlier, maybe after introducing them in line 76?

Line 94: what is an orange-shifted cell? Is the peak shifted towards shorter wavelengths, from red, or to longer wavelengths, from green? It may be good to always give a wavelength with a human colour name. This applies to other places in the manuscript as well, for instance to line 106 “further red-shifted”.

Line 111: see my comments to lines 78ff: what is a R- unit, is this a cell inhibited by a red receptor, or is it the G+R- cell tested with red light, or something else?

Line 118f: this statement remains unclear: can you see whether you recorded from receptors in ommatidia with red eyeshine, in the tested species, or do you simply correlate the existence of such ommatidia with the finding of the described cells. In other words, how does the red screening pigment contribute to the findings? Is it needed to create the red-shifted sensitivity of the R9 receptors, which likely have the same pigment as R5-8?

Line 123 Red postman is *Heliconius erato*, please add species name.

Line 125: you contradict yourself here: in line 89 you report that in some species, the G+ was blue-sensitive. Here you say you never found that. Please clarify.

Review form: Reviewer 2

Recommendation

Major revision is needed (please make suggestions in comments)

Scientific importance: Is the manuscript an original and important contribution to its field?

Excellent

General interest: Is the paper of sufficient general interest?

Good

Quality of the paper: Is the overall quality of the paper suitable?

Good

Is the length of the paper justified?

Yes

Should the paper be seen by a specialist statistical reviewer?

No

Do you have any concerns about statistical analyses in this paper? If so, please specify them explicitly in your report.

No

It is a condition of publication that authors make their supporting data, code and materials available - either as supplementary material or hosted in an external repository. Please rate, if applicable, the supporting data on the following criteria.

Is it accessible?

No

Is it clear?

No

Is it adequate?

No

Do you have any ethical concerns with this paper?

No

Comments to the Author

This study examines a newly discovered green-sensitive photoreceptor cell type across nymphalid butterflies. These cells are found in place of typically UV or blue color-sensing photoreceptor cells and hyperpolarize in the red range. The authors characterize the spectral and polarization sensitivity of these cells as well as some basic neurophysiological properties under dark, red, and green adaptation. The implication is that light passes through distal rhodopsin and filtering pigments, making the tiny proximal R9 cell effectively a red sensing cell. The R9 also directly synapses on these green sensitive cells, forming a direct opponent mechanism at the photoreceptor cell level. Although the R9 cell is not directly measured, and the synapse is not identified through imaging or other means, the authors do multiple correlative experiments to show this relationship is likely real. The data as presented are convincing that this cell type and inhibitory relationship exists, and that it is seen across the family.

Although the study appears sound, and is potentially an exciting result, a lack of clarity of the species and replicates used per method and which data the authors use to generate their results makes it difficult to fully assess their conclusions across the whole family Nymphalidae. It appears the authors base some conclusions on a single data point or two, in a single species or two, with no backup data in the supplement. I have a few major organizational issues, which, if addressed should make the manuscript and its conclusions much stronger.

Major Points

1. It is difficult to fully assess all the authors' conclusions especially in a comparative way, because there are some issues with data organization and reporting. It is unclear how many replicates were done in each species for each method (staining of photoreceptor cells, spectral characterization, polarization characterization, eye shine, spatial simulations, voltage response delays), and also whether the modeled sensitivity of G+R- is applicable to all species (presumably with distinct spectral sensitivities and eye sizes). It would be helpful to clearly include numbers of replicates and species examined for each method either in the methods or in a table, and also to have clear supplementary figures showing examples of these data, or averages of multiple replicates, on which the main figures and conclusions are based. Further, if there are differences between species these should be discussed.
2. The paper is organized a bit confusingly, and it would help to have clearer demarcations between each measurement and rationales for why each was done and why the result matters/how does it relate to other results throughout the paper. For instance, Figure 4, which is not referred to until the discussion, seems like it would be better suited as part of Figure 1 or just after it, as this describes the presence of this cell type in all these species. Also Fig. 3g is the concluding model summarizing all the previous findings, so it feels strange to reintroduce figure 4 after the conclusion has been made. Figure 3 has multiple somewhat related measures of these cell types, but it's not clear how they all go together in a single figure or even which species these were done in. This is also reflected in the text, which bounces between figure 2 and 3 and multiple panels without really introducing why or connecting multiple lines of evidence (lines

72-99).

In general, I think the findings of the paper are exciting and significant. A bit more organization and detail expanding on the implications of the results will both help solidify the conclusions and make the manuscript more accessible to a broader readership.

Minor Points:

line 36: "Physiological evidence for red-sensitive photoreceptors..." There is evidence for a 620nm cell and a 700nm interneuron in old recordings of *H. erato*. Swihart's work in the 1960s and 1970s, especially:

S. L. Swihart, W. C. Gordon, Red photoreceptor in butterflies. *Nature* 231, 126-127 (1971).

S. L. Swihart The neural basis of colour vision in the butterfly, *Papilio troilus*. *J. Insect Physiol.* 16, 1623-1636 (1970).

line 52: "It is not yet known..." Does work from the Arikawa lab or in *Drosophila* give any context about R3-8 signaling to the medulla?

line 61-62: "the retina contains green and red-sensitive photoreceptors with long visual fibres that together build opponent pairs..." Do we know this is always the case? What about the R9 in ommatidia without red filtering pigment? Is it known where these cells are synapsing, in the lamina or medulla or multiple spots along the axon? How do we know other cells are not involved?

this type of opponency seems to be like *Drosophila* R7/R8 except that butterflies essentially have two R7s, each on top of the R9 cell, how is this similar or different?

Line 72: "When recording from the axons..." How do you know you are recording from axons? It's not clear from the methods how you distinguish distal vs. proximal retina, axon, or downstream neurons in the lamina.

Line 76: Citation number 5, seems like the wrong citation for this sentence.

Line 84: "The magnitude (Ψ) and angular maximum (Φ) of polarization..." Part of what I mentioned before, the results are launched into here for pol sensitivity without any intro or rationale, and are not summarized for context. Tell the reader what is the significance of these recordings to the overall story.

Line 89: What does "equivalent" mean here?

Line 93: "Within several hundred cells" What are the cells you recorded from? How many from each species? With which method? Did you dye inject all of these? How many were injected? How many were measured for pol sensitivity, reversal potential, etc etc?

Line 109: "Importantly, the red pigment significantly reduces..." What does this suggest about the existence of a cell pair without the red pigment? Did you find this type of G+Y- unit? Do you predict it might exist?

Line 112: "The reduced sensitivity overlap..." I don't think I understand this sentence. Please elaborate.

Line 126: "R9 thus seems to have a specific role..." If this is true, why is R9 found in every ommatidium, even those without LWRh expressed in R1/R2? What is it doing there? Is there ever a case in *Vanessa* or other non-red pigmented eyes, where LWRh is expressed in R1/R2

cells? What about opponency in downstream neurons rather than the photoreceptor cell level?

Line 161: Do you mean Vanessa atalanta?

Line 176: "Opponent signals..." Could you explain a bit more your animal setup? How were the animals immobilized? How were cells exposed/made available for recording? How do you know where on the proximal-distal axis in the retina you were recording from? Did you do dye injections every time? While recording in the lamina, how could you be sure you were in the axon of a photoreceptor cell or recording from opponent lamina cells?

Line 187: "To visualize the impaled cells..." How many cells were recorded in this way versus above? How does using different microelectrode tip size change the sensitivity analyses?

Figures and Legends:

Figure 1:

b) Did you stain for opsin expression in dye injected retinula cells? How many did you identify via imaging this way and in which species?

d) please explain the red and blue triangles for current injection. Where was this measurement taken, axon or proximal retina? How many replicates and species was this done for?

Figure 2:

a) What species is this taken from? Why are the numbers different for dark, green, and red adapted cells? Was each cell only measured under a single adaptation state?

b) where is the evidence this is a diagonal cell? Why are some cells averaged while others are single?

c) These sensitivities are quite different from monarch in (b). How do you reconcile these differences? I imagine if you show the sensitivities from multiple species they are all different. Can you show the sensitivities of the other species in supplement, and have averages or single cells for all, or at least an explanation of how you dealt with these differences between species? In legend: what does "Indem" mean?

d) Is this supposed to be a model for all nymphalids? Given the differences in spectral sensitivities in b and c, and eye size, differences in opsin expression across retina, etc. I'm not sure how generalizable this model is? Could you please explain how you are using it/generalizing from it.

e and f) Where is the histology/eyeshine for the other species? Either need to include in supplement or include references from which you used eyeshine data for each species in supplement or combination of both.

Figure 3

a - d) Again I do not know which species these results come from or how many replicates?

f) Did you do this more than once, or in more than one species?

Figure 4

This is the evidence of the G+R- cell type in each species. Shouldn't this be presented first? Then move into more detailed analyses in the other figures in a subset of species or replicates. Also are

these each from a single cell? Are there replicates (in supplement)?

Table 1:

Why are there two G+ sensitivities for *Danaus* and *Heliconius*? What does this mean? Please define FWHM. The number of G+R- cells recorded is listed, with 0 in species with no red filtering pigment. Out of how many recordings total? Especially in *Vanessa*, proof of absence is harder to show, how many green cells were recorded from and did not display this inhibitory dynamic? Are there common themes between species/common ration of G+R- to normal G, B, or UV cells?

Supplement: Most of the supplement was in file types that were not accessible.

Decision letter (RSPB-2021-1560.R0)

07-Aug-2021

Dear Dr Pirih:

Your manuscript has now been peer reviewed and the reviews have been assessed by an Associate Editor. The reviewers' comments (not including confidential comments to the Editor) and the comments from the Associate Editor are included at the end of this email for your reference. As you will see, the reviewers and the Editors have raised some concerns with your manuscript and we would like to invite you to revise your manuscript to address them.

Research ethics:

Use of animals and field studies:

It is a condition of publication that you make available the data and research materials supporting the results in the article. Please see our Data Sharing Policies (<https://royalsociety.org/journals/authors/author-guidelines/#data>). Datasets should be deposited in an appropriate publicly available repository and details of the associated accession number, link or DOI to the datasets must be included in the Data Accessibility section of the article (<https://royalsociety.org/journals/ethics-policies/data-sharing-mining/>). Reference(s) to datasets should also be included in the reference list of the article with DOIs (where available).

Please submit a copy of your revised paper within three weeks. If we do not hear from you within this time your manuscript will be rejected. If you are unable to meet this deadline please let us know as soon as possible, as we may be able to grant a short extension.

Best wishes,

Dr Sasha Dall

Associate Editor
 Board Member: 1
 Comments to Author:
 Associate Editor: Doug Altshuler

Pirih et al. present interesting and exciting results on colour encoding by insect photoreceptors and help solve a mystery about insect p9. The figures are appealing, and have a strong narrative that matches the paper. We obtained two excellent reviews. Although both referees are enthusiastic about the data set and analysis, they brought up a number of substantial concerns about presentation and clarity of the manuscript. These do not seem to relate to the science so no further data should be required. However, it would be helpful to see if the authors can address these concerns in a revised manuscript. In my opinion, things would work best if they could follow the typical systems neuroscience approach of moving from raw to analyzed representative data, followed by analyzed population data. That approach would address much of the second referee's concerns.

Reviewer(s)' Comments to Author:

Referee: 1

Comments to the Author(s)

This manuscript clarifies a long-standing important question in insect colour vision: the potential role of the mystic photoreceptor number 9. The authors investigated this in several species of Nymphalid butterflies, but it is likely relevant beyond this group. The methods are all sound and the results clear, but in some parts, the description is a bit too short for a general reader to understand all details and the impact of them.

This is mainly a result of the short format, but could also be helped by clearer definitions of the exact receptor types and units in focus. I have made a few suggestions for such clarifications, all very minor.

I also think it would be nice to cite the earliest (and since then, often disputed) finding of electrical inhibition in butterfly photoreceptor signals: Matic T (1983) Electrical inhibition in the retina of the butterfly *Papilio I.* Four types of photoreceptors. *J Comp Physiol* 152: 169-82, and mention the more recent findings of mutual photoreceptor inhibition in the fly retina, there are various recent papers on *Drosophila*.

Else, I think this is a very important contribution to the field, congratulations to the authors.

Minor comments:

Title and elsewhere: not everybody knows that the wonderful name brushfoot butterflies refers to Nymphalidae, maybe good to mention it as "(Nymphalidae)" in the title

Line 53: replace "how does it" with "how it does"

Line 55f: it would be good to better relate the sentence on filtering with the "optical filtering" in the sentence before, maybe by moving "optical filtering" last in the list.

Lines 67-77: this is a very short summary of results, hard to digest for a non-specialist reader. For instance, while it can be inferred from the fact that red light induces hyperpolarization, this is not clear from a non-specialist reader, so "the inhibition" mentioned in line 76 may need to be introduced more explicitly. A few additional points that could be clarified:

Line 67: "measurements" sounds very general, maybe good to add "electrophysiological" to clarify

Line 68: "spectral stimulation" means stimulation with spectral light" – maybe good to say, for the non-specialist reader

Line 77: not clear what a "receivingfrom colour-opponent, red-sensitive unit" means, does it rather mean: receiving colour-opponent...from red-sensitive units"?

Line 78ff: you have just defined “R+G-“ class of receptors, now you introduce “G+ and R- units“ – maybe better to use consistent naming? Are you now talking about one cell type now, or about two different units that give input to the G+R- cells?

Line 88: to clarify which cells you record from, is it possible to give the information that G+R-calls are R1/2 a bit earlier, maybe after introducing them in line 76?

Line 94: what is an orange-shifted cell? Is the peak shifted towards shorter wavelengths, from red, or to longer wavelengths, from green? It may be good to always give a wavelength with a human colour name. This applies to other places in the manuscript as well, for instance to line 106 “further red-shifted”.

Line 111: see my comments to lines 78ff: what is a R- unit, is this a cell inhibited by a red receptor, or is it the G+R- cell tested with red light, or something else?

Line 118f: this statement remains unclear: can you see whether you recorded from receptors in ommatidia with red eyeshine, in the tested species, or do you simply correlate the existence of such ommatidia with the finding of the described cells. In other words, how does the red screening pigment contribute to the findings? Is it needed to create the red-shifted sensitivity of the R9 receptors, which likely have the same pigment as R5-8?

Line 123 Red postman is *Heliconius erato*, please add species name.

Line 125: you contradict yourself here: in line 89 you report that in some species, the G+ was blue-sensitive. Here you say you never found that. Please clarify.

Referee: 2

Comments to the Author(s)

This study examines a newly discovered green-sensitive photoreceptor cell type across nymphalid butterflies. These cells are found in place of typically UV or blue color-sensing photoreceptor cells and hyperpolarize in the red range. The authors characterize the spectral and polarization sensitivity of these cells as well as some basic neurophysiological properties under dark, red, and green adaptation. The implication is that light passes through distal rhodopsin and filtering pigments, making the tiny proximal R9 cell effectively a red sensing cell. The R9 also directly synapses on these green sensitive cells, forming a direct opponent mechanism at the photoreceptor cell level. Although the R9 cell is not directly measured, and the synapse is not identified through imaging or other means, the authors do multiple correlative experiments to show this relationship is likely real. The data as presented are convincing that this cell type and inhibitory relationship exists, and that it is seen across the family.

Although the study appears sound, and is potentially an exciting result, a lack of clarity of the species and replicates used per method and which data the authors use to generate their results makes it difficult to fully assess their conclusions across the whole family Nymphalidae. It appears the authors base some conclusions on a single data point or two, in a single species or two, with no backup data in the supplement. I have a few major organizational issues, which, if addressed should make the manuscript and its conclusions much stronger.

Major Points

1. It is difficult to fully assess all the authors’ conclusions especially in a comparative way, because there are some issues with data organization and reporting. It is unclear how many replicates were done in each species for each method (staining of photoreceptor cells, spectral characterization, polarization characterization, eye shine, spatial simulations, voltage response delays), and also whether the modeled sensitivity of G+R- is applicable to all species (presumably with distinct spectral sensitivities and eye sizes). It would be helpful to clearly include numbers of replicates and species examined for each method either in the methods or in a table, and also to have clear supplementary figures showing examples of these data, or averages of multiple

replicates, on which the main figures and conclusions are based. Further, if there are differences between species these should be discussed.

2. The paper is organized a bit confusingly, and it would help to have clearer demarcations between each measurement and rationales for why each was done and why the result matters/how does it relate to other results throughout the paper. For instance, Figure 4, which is not referred to until the discussion, seems like it would be better suited as part of Figure 1 or just after it, as this describes the presence of this cell type in all these species. Also Fig. 3g is the concluding model summarizing all the previous findings, so it feels strange to reintroduce figure 4 after the conclusion has been made. Figure 3 has multiple somewhat related measures of these cell types, but it's not clear how they all go together in a single figure or even which species these were done in. This is also reflected in the text, which bounces between figure 2 and 3 and multiple panels without really introducing why or connecting multiple lines of evidence (lines 72-99).

In general, I think the findings of the paper are exciting and significant. A bit more organization and detail expanding on the implications of the results will both help solidify the conclusions and make the manuscript more accessible to a broader readership.

Minor Points:

line 36: "Physiological evidence for red-sensitive photoreceptors..." There is evidence for a 620nm cell and a 700nm interneuron in old recordings of *H. erato*. Swihart's work in the 1960s and 1970s, especially:

S. L. Swihart, W. C. Gordon, Red photoreceptor in butterflies. *Nature* 231, 126-127 (1971).

S. L. Swihart The neural basis of colour vision in the butterfly, *Papilio troilus*. *J. Insect Physiol.* 16, 1623-1636 (1970).

line 52: "It is not yet known..." Does work from the Arikawa lab or in *Drosophila* give any context about R3-8 signaling to the medulla?

line 61-62: "the retina contains green and red-sensitive photoreceptors with long visual fibres that together build opponent pairs..." Do we know this is always the case? What about the R9 in ommatidia without red filtering pigment? Is it known where these cells are synapsing, in the lamina or medulla or multiple spots along the axon? How do we know other cells are not involved?

this type of opponency seems to be like *Drosophila* R7/R8 except that butterflies essentially have two R7s, each on top of the R9 cell, how is this similar or different?

Line 72: "When recording from the axons..." How do you know you are recording from axons? It's not clear from the methods how you distinguish distal vs. proximal retina, axon, or downstream neurons in the lamina.

Line 76: Citation number 5, seems like the wrong citation for this sentence.

Line 84: "The magnitude (Ψ) and angular maximum (Φ) of polarization..." Part of what I mentioned before, the results are launched into here for pol sensitivity without any intro or rationale, and are not summarized for context. Tell the reader what is the significance of these recordings to the overall story.

Line 89: What does "equivalent" mean here?

Line 93: "Within several hundred cells" What are the cells you recorded from? How many from each species? With which method? Did you dye inject all of these? How many were injected? How many were measured for pol sensitivity, reversal potential, etc etc?

Line 109: "Importantly, the red pigment significantly reduces..." What does this suggest about the existence of a cell pair without the red pigment? Did you find this type of G+Y- unit? Do you predict it might exist?

Line 112: "The reduced sensitivity overlap..." I don't think I understand this sentence. Please elaborate.

Line 126: "R9 thus seems to have a specific role..." If this is true, why is R9 found in every ommatidium, even those without LWRh expressed in R1/R2? What is it doing there? Is there ever a case in Vanessa or other non-red pigmented eyes, where LWRh is expressed in R1/R2 cells? What about opponency in downstream neurons rather than the photoreceptor cell level?

Line 161: Do you mean Vanessa atalanta?

Line 176: "Opponent signals..." Could you explain a bit more your animal setup? How were the animals immobilized? How were cells exposed/made available for recording? How do you know where on the proximal-distal axis in the retina you were recording from? Did you do dye injections every time? While recording in the lamina, how could you be sure you were in the axon of a photoreceptor cell or recording from opponent lamina cells?

Line 187: "To visualize the impaled cells..." How many cells were recorded in this way versus above? How does using different microelectrode tip size change the sensitivity analyses?

Figures and Legends:

Figure 1:

b) Did you stain for opsin expression in dye injected retinula cells? How many did you identify via imaging this way and in which species?

d) please explain the red and blue triangles for current injection. Where was this measurement taken, axon or proximal retina? How many replicates and species was this done for?

Figure 2:

a) What species is this taken from? Why are the numbers different for dark, green, and red adapted cells? Was each cell only measured under a single adaptation state?

b) where is the evidence this is a diagonal cell? Why are some cells averaged while others are single?

c) These sensitivities are quite different from monarch in (b). How do you reconcile these differences? I imagine if you show the sensitivities from multiple species they are all different. Can you show the sensitivities of the other species in supplement, and have averages or single cells for all, or at least an explanation of how you dealt with these differences between species? In legend: what does "Indem" mean?

d) Is this supposed to be a model for all nymphalids? Given the differences in spectral sensitivities in b and c, and eye size, differences in opsin expression across retina, etc. I'm not sure how generalizable this model is? Could you please explain how you are using it/generalizing from it.

e and f) Where is the histology/eyeshine for the other species? Either need to include in supplement or include references from which you used eyeshine data for each species in supplement or combination of both.

Figure 3

a - d) Again I do not know which species these results come from or how many replicates?

f) Did you do this more than once, or in more than one species?

Figure 4

This is the evidence of the G+R- cell type in each species. Shouldn't this be presented first? Then move into more detailed analyses in the other figures in a subset of species or replicates. Also are these each from a single cell? Are there replicates (in supplement)?

Table 1:

Why are there two G+ sensitivities for *Danaus* and *Heliconius*? What does this mean? Please define FWHM. The number of G+R- cells recorded is listed, with 0 in species with no red filtering pigment. Out of how many recordings total? Especially in *Vanessa*, proof of absence is harder to show, how many green cells were recorded from and did not display this inhibitory dynamic? Are there common themes between species/common ration of G+R- to normal G, B, or UV cells?

Supplement: Most of the supplement was in file types that were not accessible.

Author's Response to Decision Letter for (RSPB-2021-1560.R0)

See Appendix A.

Decision letter (RSPB-2021-1560.R1)

01-Oct-2021

Dear Dr Pirih

I am pleased to inform you that your manuscript entitled "Red-green opponency in the long visual fibre photoreceptors of brushfoot butterflies (Nymphalidae)" has been accepted for publication in *Proceedings B*.

Data Accessibility section

Open Access

Paper charges

Sincerely,

Dr Sasha Dall

Associate Editor:

Comments to Author:

Associate Editor: Doug Altshuler

Pirih et al. have submitted a revised manuscript that is fully responsive to the great suggestions from the referees. This is a a fascinating study the physiology of insect photoreceptors, and it should have broad appeal to our readership.

Appendix A

Pirih et al. present interesting and exciting results on colour encoding by insect photoreceptors and help solve a mystery about insect **R9**. The figures are appealing, and have a strong narrative that matches the paper. We obtained two excellent reviews. Although both referees are enthusiastic about the data set and analysis, they brought up a number of substantial concerns about presentation and clarity of the manuscript. These do not seem to relate to the science so no further data should be required. However, it would be helpful to see if the authors can address these concerns in a revised manuscript. **In my opinion, things would work best if they could follow the typical systems neuroscience approach of moving from raw to analyzed representative data, followed by analyzed population data.** That approach would address much of the second referee's concerns.

We are grateful to the reviewers for the two insightful reviews with many helpful comments. We have addressed all the concerns. Particularly, we have joined the data from the former Figure 4 to Figure 1, and created a six page Supplement with one supplementary table and five supplementary figures.

Referee 1

This manuscript clarifies a long-standing important question in insect colour vision: the potential role of the mystic photoreceptor number 9. The authors investigated this in several species of Nymphalid butterflies, but it is likely relevant beyond this group. The methods are all sound and the results clear, but in some parts, **the description is a bit too short** for a general reader to understand all details and the impact of them. This is mainly a result of the short format, but could also be helped by clearer definitions of the exact receptor types and units in focus. I have made a few suggestions for such clarifications, all very minor.

Indeed the manuscript was still recovering from the “brief communication”. We are happy to have had the chance to expand the material and we hope that the revision is now suitable, also in terms of nomenclature.

I also think it would be nice to cite the earliest (and since them, often disputed) finding electrical inhibition in butterfly photoreceptor signals : **Matic T (1983)** Electrical inhibition in the retina of the butterfly *Papilio I*. Four types of photoreceptors. *J Comp Physiol* 152: 169-82, and mention the more **recent findings** of mutual photoreceptor inhibition **in the fly retina**, there are various recent papers on *Drosophila*. Else, I think this is a very important contribution to the field, congratulation to the authors.

Thank you for the kindest words. We have added the reference to T Matic but we note that hyperpolarisations in *Papilio*, which have been ascribed to an extracellular action in the paper of Matic, have been in the meantime shown to be synaptic, but between vertical R1/2 and proximal R5-8, not the basal R9.

As the references to *Drosophila* work, we have already cited [Heath ... et Behnia 2020] and we add [Schnaitman 2018].

Minor comments

1. Title and elsewhere: not everybody knows that the wonderful name brushfoot butterflies refers to Nymphalidae, maybe good to mention it as “(Nymphalidae)” in the title.

We have changed the title.

2. Line 53: replace “how does it” with “how it does”.

Done.

3. Line 55f: it would be good to better relate the sentence on filtering with the “optical filtering” in the sentence before, maybe by moving “optical filtering last in the list.

Order changed.

4. Lines 67-77: this is a very short summary of results, hard to digest for a non-specialist reader. For instance, while it can be inferred from the fact that red light induces hyperpolarization, this is not clear for a non-specialist reader, so “the inhibition” mentioned in line 76 may need to be introduced more explicitly.

We have expanded the paragraph with a few explanations. We added “We assumed that the hyperpolarisations were caused by inhibitory synapses from red-sensitive photoreceptors”. We changed “inhibition” to “hyperpolarisation”.

5. Line 67: “measurements” sounds very general, maybe good to add “electrophysiological” to clarify.

We changed to “intracellular electrophysiological measurements”.

6. Line 68: “spectral stimulation” means stimulation with spectral light” – maybe good to say, for the non-specialists reader.

We introduce the term “narrowband spectral light”.

7. Line 77: not clear what a “receivingfrom colour-opponent, red-sensitive unit” means, does it rather mean: receiving colour-opponent...from red-sensitive units”?
8. Line 78ff: you have just defined “R+G-“ class of receptors, now you introduce “G+ and R- units“ – maybe better to use consistent naming? Are you now talking about one cell type now, or about two different units that give input to the G+R- cells?
9. Line 88: to clarify which cells you record from, is it possible to give the information that G+R- calls are R1/2 a bit earlier, maybe after introducing them in line 76?

We changed the last part of the paragraph to: “We termed the novel photoreceptor class G+R-: **green-sensitive cells (G+)**, inhibited by postsynaptic currents from **red-sensitive units (R-)**. Below, we provide the evidence that the G+R- cells are R1&2 and their R- units are likely R9. The novel G+R- photoreceptor class is the retinal basis for green-red colour opponency.”

We hope that we have made the terminology consistent. The idea behind the terminology is the following: “R- unit” is meant as the physiological entity that is responsible for the hyperpolarisation of the green R12. We then show that this is the red R9 receptor.

10. Line 94: what is an orange-shifted cell? Is the peak shifted towards shorter wavelengths, from red, or to longer wavelengths, from green? It may be good to always give a wavelength with a human colour name. This applies to other places in the manuscript as well, for instance to line 106 “further red-shifted”.

We have removed the cosmological term. The sentence in question is now: “Within 487 recorded cells (Table S1), we found all common photoreceptor classes (UV, B, G), including green-sensitive cells with spectral sensitivity maxima shifted to orange”, and to “The sensitivity peak of R9 is further long-wavelength shifted to red by the screening pigment.”

11. Line 111: see my comments to lines 78ff: what is a R- unit, is this a cell inhibited by a red receptor, or is it the G+R- cell tested with red light, or something else?

We hope all is fine now, see our response at point 7-9.

12. Line 118f: this statement remains unclear: can you see whether you recorded from receptors in ommatidia with red eyeshine, in the tested species, or do you simply correlate the existence of such ommatidia with the finding of the described cells. In other words, how does the red

screening pigment contribute to the findings? Is it needed to create the red-shifted sensitivity of the R9 receptors, which likely have the same (visual) pigment as R5-8?

Changed and expanded to: “The presence of ommatidia with red screening pigments is not only correlated by the finding of G+R- cells, but seems to be a requirement for the red-shifted sensitivity of R9 and hence the implementation of the extended colour discrimination range.”

13. Line 123 Red postman is *Heliconius erato*, please add species name.

Done.

14. Line 125: you contradict yourself here: in line 89 you report that in some species, the G+ was blue-sensitive. Here you say you never found that. Please clarify.

This comment refers to the data presented previously in Fig 2bc, and now also in Fig S2. G+ unit of R1/2 always had a peak in the cyan-green (500~530 nm), and in *Danaus*, *Heliconius* & *Melitaea*, an additional peak in the blue (~450 nm). “We never found a blue cell” referred to a cell with pure, single-template blue peak that would exhibit red inhibition. The corresponding text is now:

“In *D. plexippus* and *H. erato*, some G+R- cells had broadened sensitivity with two maxima, in the green and blue (Fig. 2b,c), in line with the recently found co-expression of blue and LW opsins in *Heliconius* R1&2. Varying levels of opsin co-expression may be the main cause for the different spectral sensitivities of G+R- cells in different species (Fig. S2).”

Referee: 2

This study examines a newly discovered green-sensitive photoreceptor cell type across nymphalid butterflies. These cells are found in place of typically UV or blue color-sensing photoreceptor cells and hyperpolarize in the red range. The authors characterize the spectral and polarization sensitivity of these cells as well as some basic neurophysiological properties under dark, red, and green adaptation. The implication is that light passes through distal rhodopsin and filtering pigments, making the tiny proximal R9 cell effectively a red sensing cell. The R9 also directly synapses on these green sensitive cells, forming a direct opponent mechanism at the photoreceptor cell level. Although the R9 cell is not directly measured, and the synapse is not identified through imaging or other means, the authors do **multiple correlative experiments** to show this relationship is likely real. The data as presented are convincing that this cell type and inhibitory relationship exists, and that it is seen across the family.

Although the study appears sound, and is potentially an exciting result, a lack of **clarity of the species and replicates** used per method and which data the authors use to generate their results makes it difficult to fully assess their conclusions across the whole family Nymphalidae. It appears the authors base some conclusions on **a single data point or two**, in a single species or two, with no backup data in the supplement. I have a few major **organizational issues**, which, if addressed should make the manuscript and its conclusions much stronger.

We are grateful for the comments, we have gambatted to do our best addressing them. Particularly, the “single datapoint” issue was due to the previous attempt as a brief communication. We have used the comments as the motivation to expand the data set now presented in the Supplement.

Major Points

It is difficult to fully assess all the authors' conclusions especially in a comparative way, because there are some issues with data organization and reporting. It is unclear how many replicates were done in each species for each method (staining of photoreceptor cells, spectral characterization, polarization characterization, eye shine, spatial simulations, voltage response delays), and also whether the modeled sensitivity of G+R- is applicable to all species (presumably with distinct spectral sensitivities and eye sizes). It would be helpful to clearly include numbers of replicates and species examined for each method either in the methods or in **a table**, and also to have clear supplementary figures showing examples of these data, or averages of multiple replicates, on which the main figures and conclusions are based. **Further, if there are differences between species these should be discussed.**

The newly created Supplement now starts with a table S1 where the fate of the 478 measured cells is given. We have characterised 52 G+R- cells via spectral and polarisation sensitivity measurements, 11-26 were subjected to more additional protocols. The main text contains an abridged table. The differences in the sensitivity of G+ unit are now briefly explained as:

“In *D. plexippus* and *H. erato*, some G+R- cells had broadened sensitivity with two maxima, in the green and blue (Fig. 2b,c), in line with the recently found co-expression of blue and LW opsins in *Heliconius* R1&2. Varying levels of opsin co-expression may be the main cause for the different spectral sensitivities of G+R- cells in different species (Fig. S2).”

The paper is organized a bit confusingly, and it would help to have clearer demarcations between each measurement and rationales for why each was done and why the result matters/how does it relate to other results throughout the paper. For instance, Figure 4, which is not referred to until the discussion, seems like it would be better suited as part of Figure 1 or just after it, as this describes the presence of this cell type in all these species. Also Fig. 3g is the concluding model summarizing all the previous findings, so it feels strange to reintroduce figure 4 after the conclusion has been made. This is also reflected in the text, which bounces between figure 2 and 3 and multiple panels without really introducing why or connecting multiple lines of evidence (lines 72-99).

Thank you very much for this suggestion. Figure 4 has been joined to Figure 1, which should solve the issue. The rationale for measuring polarisation sensitivity is explained in Results: “Response to a rotating polariser was measured to estimate the angular maximum of polarisation sensitivity, which coincides with the microvillar orientation, indicative of the receptor position within the ommatidium.” with a reference to [Lin 1993]

Figure 3 has multiple somewhat related measures of these cell types, but it's not clear how they all go together in a single figure or even which species these were done in.

The names of species are shown in italics: (a,b,c) is *Archaeoprepona*, while (d) is the cells from *Danaus* and *Heliconius*, whose spectral sensitivities are reported in Figure 2bc.

In general, I think the findings of the paper are exciting and significant. A bit more organization and detail expanding on the implications of the results will both help solidify the conclusions and make the manuscript more accessible to a broader readership.

Thank you for the kindest words and the motivation for showing more data, we indeed hope that the paper (and the new Supplement!) will gain a substantial readership.

Minor Points

line 36: “Physiological evidence for red-sensitive photoreceptors...” There is evidence for a 620nm cell and a 700nm interneuron in old recordings of *H. erato*. Swihart’s work in the 1960s and 1970s, especially: S. L. Swihart, W. C. Gordon, Red photoreceptor in butterflies. *Nature* 231, 126–127 (1971), and S. L. Swihart, The neural basis of colour vision in the butterfly, *Papilio troilus*. *J. Insect Physiol.* 16, 1623–1636 (1970).

We have read the Heliconius work of Swihart et al. which does bear some historical significance. For Swihart and Gordon, we opine that the ERG method there is inaccurate and prone to adaptation artefacts. We included a citation to [Swihart 1972] reporting colour-selective interneurons.

line 52: “It is not yet known...” Does work from the Arikawa lab or in *Drosophila* give any context about R3-8 signaling to the medulla?

The sentence refers to the problem of conveying long wavelength information to the medulla. In *Drosophila*, this is solved by the green-sensitive long visual fibre R8y, while the svfs coming from broadband outer receptors (R1-6) terminate in the lamina, which is in Diptera specialised to function in neural superposition.

In *Papilio*, green-sensitive long visual fibres do not exist: R1 and R2 can be only UV, blue or violet-sensitive. Short visual fibres R3-4 are green-sensitive in all ommatidia and R5-8 are either green-, broadband- or red-sensitive. The only currently known pathway connecting R3-8 and medulla is going through LMC neurons. However, the spectral sensitivity recorded from virtually all LMC neurones was very broad, in contrast with the narrow sensitivity of red-sensitive R5-8. The functioning of en-passant synapses of R3-8 in *Papilio* has not been sufficiently determined. We must therefore consider the problem of downstream processing of long wavelength information in butterflies (both Papilionids and Nymphalids) as unsolved. We have included the following sentence in the Intro:

“It is not yet known how the neural signal is conveyed from R3-8 to the medulla, and how it does contribute to colour vision. In *Papilio*, for instance, the only known neural pathway, connecting the long-wavelength sensitive R3-8 and medulla, are the laminar monopolar cells, neurones with very broad spectral sensitivities.”

line 61-62: “the retina contains green and red-sensitive photoreceptors with long visual fibres that together build opponent pairs...” Do we know this is always the case? What about the R9 in ommatidia without red filtering pigment? Is it known where these cells are synapsing, in the lamina or medulla or multiple spots along the axon? How do we know other cells are not involved? this type of opponency seems to be like *Drosophila* R7/R8 except that butterflies essentially have two R7s, each on top of the R9 cell, how is this similar or different?

Thank you for this most intriguing series of questions. Unfortunately, to our knowledge, the Nymphalids are a completely uncharted territory for almost all of these questions. In both butterfly families with a fully tiered retina, R9 is a long visual fibre (*Papilio*: Arikawa; *Pieris*: Shimohigashi and Tominaga). Answering the question about the function of the R9 long visual fibre in the Nymphalid and Lycaenid ommatidia without the red screening pigment will likely require a thorough study of ultrastructure and connectomics.

With respect to *Drosophila*, R7 and R8 have direct synapses, and this is indeed homologous to R1 vs R9 in brushfoots. However, a direct synapse between the twin sisters R1 and R2 (autosynapse R7 to R7?) does not seem to exist. We will reflect upon these ideas our next work, but will refrain from becoming too philosophical in the current manuscript.

We share your pain in the Discussion: “R9 thus seems to have a specific role in the red ommatidia, in providing antagonistic input only to the green-sensitive R1/2 photoreceptors, while its role in the non-red ommatidia remains unknown. Still, the long fibre of R9 may play a more general role in colour processing in the medulla.”

Line 72: “When recording from the axons...” How do you know you are recording from axons? It’s not clear from the methods how you distinguish distal vs. proximal retina, axon, or downstream neurons in the lamina.

We have not recorded from the downstream neurons. We had good control of the electrode position (described in Methods). When being proximally, i.e. close to the fenestrated layer, as indicated by frequent loss of contact due to the tip of the electrode being in the tracheolae, the potentials recorded from R1&2 were prone to spiking, indicating an active propagation mechanism being present in the narrow axons of photoreceptors R1&2.

Line 76: Citation number 5, seems like the wrong citation for this sentence.

Indeed, 5 should be 15, corrected to “Chen, Belušič & Arikawa”.

Line 84: “The magnitude (Ψ) and angular maximum (Φ) of polarization...” Part of what I mentioned before, the results are launched into here for pol sensitivity without any intro or rationale, and are not summarized for context. Tell the reader what is the significance of these recordings to the overall story.

We added an introductory sentence: “Response to a rotating polariser was measured to estimate the angular maximum of polarisation sensitivity, which coincides with the microvillar orientation, indicative of the receptor position within the ommatidium” [Lin].

Line 89: What does “equivalent” mean here?

We have changed to “some”. Now it reads: “Microelectrode dye injection confirmed that G+R- cells were indeed R1/2 (Fig. 1b). In *D. plexippus* and *H. erato*, **some G+R- R1/2** cells had broadened sensitivity with two maxima, in the green and blue (Fig. 2b,c), in line with the recently found co-expression of blue and LW opsins in *Heliconius* R1&2.”

Line 93: “Within several hundred cells” What are the cells you recorded from? How many from each species? With which method? Did you dye inject all of these? How many were injected? How many were measured for pol sensitivity, reversal potential, etc etc?

We have included an expanded table with the relevant data into Supplementary material. In short: 478 cells from 70 animals were characterised in terms of spectral and polarisation sensitivity. Of those, 52 cells (**11%**) were of the G+R- type, 10 were attempted to be injected, and 2 were successful at the microscope. Between 11 and 26 G+R- cells were additionally characterised for receptive field, current clamp. We note that we have altogether skipped a great many cells (~1000?) that immediately appeared “boring” (green without hyperpolarisations) in the LED Synth sweeps, even with green selective adaptation.

Line 109: “Importantly, the red pigment significantly reduces...” What does this suggest about the existence of a cell pair without the red pigment? Did you find this type of G+Y- unit? Do you predict it might exist?

This is a most insightful question. We have not encountered units that could be described as “G+Y-” cells. The spectral sensitivity of putative “Y” (=R9-SP) cells is shown in the model pane (Fig. 2D) with a dotted line.

In the text, we now state: “In ommatidia without the red screening pigment, R9 should have substantial sensitivity in the green (Fig. 2d, pink curve). In our recordings, we have not encountered any signals from such units.”

Line 112: “The reduced sensitivity overlap...” I don’t think I understand this sentence. Please elaborate.

We now elaborate: “In the red ommatidia, the two photoreceptors in the G+R– opponent pair have mutually exclusive, non-overlapping spectral sensitivities: G+ are insensitive to red stimuli due to opponent signalling from R– which are in turn insensitive to green stimuli due to the red screening pigment. The reduced overlap of spectral sensitivity likely enhances colour discrimination ability.”

Line 126: “R9 thus seems to have a specific role...” If this is true, why is R9 found in every ommatidium, even those without LWRh expressed in R1/R2? What is it doing there? Is there ever a case in *Vanessa* or other non-red pigmented eyes, where LWRh is expressed in R1/R2 cells?

Assuming that “Y” R9 (in non-pigmented ommatidia) is a phototransductive cell, then its spectral sensitivity would be peaking somewhere redwards of ~560 nm, but will retain significant sensitivity in the green range. Optical measurements of the pupil response in a large part of the eye of *Vanessa indica* (Pirih et al. 2020) indicate that R1/2 can only be blue- or UV-sensitive there. This may however not be the case for other butterflies without the red screening pigment.

Since this is a speculative idea, we have refrained from adding it into the discussion.

What about opponency in downstream neurons rather than the photoreceptor cell level?

It probably exists, awaiting the future brave electrophysiologists and histologists. We state towards the end of the first discussion paragraph:

“We did not find any UV- or blue-sensitive R1/2 cells with opponent signals from red-sensitive cells. R9 thus seems to have a specific role in red ommatidia in providing antagonistic input only to the green-sensitive R1/2 photoreceptors, while its role in non-red ommatidia remains unknown. Still, the long fibre of R9 may play a more general role in colour processing in the medulla.”

Line 161: Do you mean *Vanessa atalanta*? Yes.

Line 176: “Opponent signals...” Could you explain a bit more your animal setup? How were the animals immobilized? How were cells exposed/made available for recording? How do you know where on the proximal-distal axis in the retina you were recording from? Did you do dye injections every time? While recording in the lamina, how could you be sure you were in the axon of a photoreceptor cell or recording from opponent lamina cells?

This is now explained *at length* in the Methods:

“The animals were immobilized in plastic tubes with a mixture of bees wax and resin and fixed with the head in the centre of rotation into a miniature goniometer. After the animal was pre-oriented for the recording, a small hole for the microelectrode was cut into the cornea and sealed with silicon vacuum grease. The reference electrode was a 50 µm diameter Ag/AgCl wire, mounted on the mini goniometer, inserted below the cornea of the recorded eye, to minimize the electroretinogram. The mini goniometer was then fixed to a large goniometer, which carried also the piezo driven micromanipulator (Sensapex, Oulu, Finland). Again, the eye was carefully positioned into the centre of rotation of the large goniometer. The dorso-ventral axis of the compound eye was aligned with the z-axis of the recording microelectrode, yielding a maximal number of cell impalements and rendering all parts of the eye accessible for the recording,

including the extreme dorsal and ventral regions. Location of the microelectrode tip during the penetration was determined with the micrometer dial on the z-axis. The depth of recording was determined by the location of the hole on the cornea, the electrode angle and estimated according to the relative quantities of impaled distal receptors R1&2 vs. the proximal R3-8. The electrode trajectory was also visible in histological sections; current clamp experiments were possible min. ~250 μm proximally from the cornea. Recordings from R1&2 axons were obtained in the proximal retina, where R1&2 do not have the rhabdomeres, in the fenestrated layer below the retina or in the lamina.”

Line 187: “To visualize the impaled cells...” How many cells were recorded in this way versus above? How does using different microelectrode tip size change the sensitivity analyses (?of spectral and polarisation sensitivity)?

We have performed the dye marking procedure in 10 animals (*Argynnis*, *Danaus*, *Morpho*, *Archaeoprepona*). The whole procedure was successfully completed in 2 specimen of *Argynnis*. In the other cases, the dye could not be located in the histological sections. We note that *Argynnis* has the smallest eyes (and photoreceptors) of the four species. Possibly, the amount of dye injected with the protocol was too small to be visible in the larger eyes.

When we were using the electrode with the dye, we were unable to perform current clamp due to higher resistance, while the all the other protocols in the bridge mode (including selective adaptation) were unaffected.

Figures and Legends:

Figure 1:

Fig 1b) Did you stain for opsin expression in dye injected retinula cells? How many did you identify via imaging this way and in which species?

Two cells were successfully injected in two specimen of *Argynnis*. We do not have access to opsin antibodies so were unable to perform co-localisation.

Fig 1d) please explain the red and blue triangles for current injection. Where was this measurement taken, axon or proximal retina? How many replicates and species was this done for?

This is now explained in the legend. As a rule, current injection is more successful when recording proximally. The measurement in Fig. 1 (*Argynnis*) was indeed recorded during a more proximal impalement, in the region where the photoreceptors R1/2 thin out into axons.

We now show current clamp experiments in *Archaeoprepona*, *Heliconius*, *Charaxes*, *Morpho*, *Danaus* and *Speyeria* in the Supplement. Current injection was not successful in *Melitaea*. The bodycount of replicates and species is in the Supplementary Table.

Figure 2:

Fig 2a) What species is this taken from? Why are the numbers different for dark, green, and red adapted cells? Was each cell only measured under a single adaptation state?

The species names are indicated in the figure panels (a: *Archaeoprepona*; b: *Danaus*; c: *Heliconius*). Two cells were measured in all three conditions, four in dark- and green-adapted state, and five only in the green-adapted state. The legend now states detailed *Ns*.

Fig 2b) where is the evidence this is a diagonal cell? Why are some cells averaged while others are single?

The evidence for these two cells obtained via measurement of polarisation sensitivity is provided in Figure 3d. This is now also written in the legend. For the sake of consistency between Fig3b

and Fig3c, we have removed the average for *Danaus* in 3c and made a whole new supplementary figure (S2). The legend for Fig 2bc states “single cells”

Fig 2c) These sensitivities are quite different from monarch in (b). How do you reconcile these differences? I imagine if you show the sensitivities from multiple species they are all different. Can you show the sensitivities of the other species in supplement, and have averages or single cells for all, or at least an explanation of how you dealt with these differences between species? In legend: what does “**Idem**” mean?

We believe the main reason for different and non-template like spectral sensitivity in some species is co-expression of blue and green opsins. This mechanism has been recently shown by [McCulloch 2021, biorxiv preprint]. Post-translational modifications of rhodopsins could also be at play here, but this is a pure speculation so we refrain from discussing it. Our approach to centering the ommatidial axis is robust, so we do not think that optical effects could importantly contribute. Indeed the diversity should be shown, and we have therefore added a supplementary figure **Fig S2**.

“*Idem*” (or “id.”) is a standard latin expression for “the same”, a tad bit less well known as “etc.” “i.e.”, “et al.” but perhaps we can keep it; the language editor might tell.

Fig 2d) Is this supposed to be a model for all nymphalids? Given the differences in spectral sensitivities in b and c, and eye size, differences in opsin expression across retina, etc. I’m not sure how generalizable this model is? Could you please explain how you are using it/generalizing from it.

The model has been synthesised from the anatomy of *Parantica sita*, red pigment absorption data from *Danaus plexippus* and opsin templates of Govardovskii with the peak parameters estimated from our measurements. To our knowledge, these are the most complete datasets currently available. Notwithstanding the effects of wave optics and dichroism, the biggest unknown in the current model is the effective optical thickness of the pigment, another can of worms that might get tackled soon. In principle, the same model can be used for different nymphalids and lycaenids as soon as more anatomy data (e.g. sfbSEM) becomes available.

Fig 2ef) Where is the histology/eyeshine for the other species? Either need to include in supplement or include references from which you used eyeshine data for each species in supplement or combination of both.

We have now included eyeshine pictures in an new supplemental figure S5.

Figure 3

Fig 3a–d)

Again I do not know which species these results come from or how many replicates?

Species names are indicated in figure panels. Replicates are now in the Supplementary table.

Fig 3f) Did you do this more than once, or in more than one species?

We now add a more detailed show of synaptic delays for *Argynnis paphia* and *Archaeoprepona demophon* in the Supplement **S4**. We shall refrain from further quantitative analysis of delays at this time: a full analysis of synaptic function will require a different experimental paradigm.

Figure 4

This is the evidence of the G+R- cell type in each species. Shouldn't this be presented first? Then move into more detailed analyses in the other figures in a subset of species or replicates. Also are these each from a single cell? Are there replicates (in supplement)?

Figure 4 has been integrated into Figure 1 and indeed the flow is better, thank you for the suggestion. The replicates are counted in the Supplement **Table** and figures **S1**, **S3**.

Table 1:

Why are there two G+ sensitivities for *Danaus* and *Heliconius*? What does this mean? Please define FWHM. The number of G+R- cells recorded is listed, with 0 in species with no red filtering pigment. Out of how many recordings total? Especially in *Vanessa*, proof of absence is harder to show, how many green cells were recorded from and did not display this inhibitory dynamic? Are there common themes between species/common ration of G+R- to normal G, B, or UV cells?

The two peak wavelengths for *Danaus* and *Heliconius* are indicating the measured peaks in double-peaked cells, and are probably due to opsin coexpression or perhaps due to post-translational modification of rhodopsin.

The numbers of characterised cells are assembled in the supplementary table **S1**. A single electrode trajectory in our setup actually gives between 30 and 300 cell impalements, depending on the eye size. By virtue of using the LED Synth, we learned about the spectral sensitivity class in a few seconds after the impalement. As the limiting factor is actually the experimenter's time, our experiments had a (desired) bias towards the "interesting" cells. We have not characterised many many R3-4 and R5-8 cells that did not show hyperpolarisations; we have focussed on the cells showing hyperpolarisations instead.

In *Vanessa atalanta* ($N=2$), for instance, we have recorded from 7 non-antagonist green cells (R3-4), ignoring at least 100 others. We note that the optical (ORG) method gives an unbiased sample of a whole eye region (Pirih et al. 2020) and is a much better method to gain information about the fractions of different ommatidial types and photoreceptor classes.

Supplement. Most of the supplement was in file types that were not accessible.

The data files can be opened with Graphpad Prism available after registration at <https://www.graphpad.com/demos/> The software will first be a fully functional demo and will become a viewer after the trial period is over. We have added a Readme with instructions to Dryad.